# RoCUS: Robot Controller Understanding via Sampling

**Yilun Zhou   Serena Booth   Nadia Figueroa   Julie Shah**
MIT CSAIL
{yilun, serenabooth, nadiafig, julie_a_shah}@csail.mit.edu

https://yilunzhou.github.io/RoCUS

**Abstract:** As robots are deployed in complex situations, engineers and end users must develop a holistic understanding of their behaviors, capabilities, and limitations. Some behaviors are directly optimized by the objective function. They often include success rate, completion time or energy consumption. Other behaviors—e.g., collision avoidance, trajectory smoothness or motion legibility—are typically emergent but equally important for safe and trustworthy deployment. Designing an objective which optimizes every aspect of robot behavior is hard. In this paper, we advocate for systematic analysis of a wide array of behaviors for holistic understanding of robot controllers and, to this end, propose a framework, RoCUS, which uses Bayesian posterior sampling to find situations where the robot controller exhibits user-specified behaviors, such as highly jerky motions. We use RoCUS to analyze three controller classes (deep learning models, rapidly exploring random trees and dynamical system formulations) on two domains (2D navigation and a 7 degree-of-freedom arm reaching), and uncover insights to further our understanding of these controllers and ultimately improve their designs.

**Keywords:** Debugging and Evaluation, Algorithmic Transparency

## 1 Introduction

In 2018, after a confluence of failures, an autonomous vehicle (AV) struck and killed a pedestrian for the first time. In the run-up to this fateful event, the responsible company had reportedly been trying to improve the AV "ride experience" by emphasizing non-critical behaviors—such as the smoothness of the ride [1]. This event reflects the long-standing challenge in robotics: designing an appropriate objective which considers both safety-critical and non-critical behaviors. When crafting an objective, it is virtually impossible to proactively account for all potential controller behaviors, and some priorities may even be in conflict with one another [2]. In practice, any given robot behaviors may be specified, unspecified, or even misspecified [3], so extensive testing and evaluation is a critical component of designing and assessing robot controllers—especially those using black-box models such as deep neural networks.

A common testing procedure focuses on finding extreme and edge cases of controller failure. For example, a tester might use this procedure to find that the AV swerves very badly when encountering a farm animal while traveling at 60mph. Finding such extreme and edge cases is well-studied within both traditional software testing paradigms [4] and more recent adversarial perturbation testing methods [5]. However, we argue that an equally, if not more, important form of testing should focus on *representative* scenarios, which considers the likelihood of encountering these scenarios. For example, if this AV is going to be deployed exclusively in New York City, the above example is largely unhelpful: cars rarely travel at 60mph in the city, and are very unlikely to encounter farm animals. Instead, the tester may prefer to know that the car swerves—though not as substantively—at lower speeds when a pedestrian steps toward it. Finding representative scenarios is often overlooked, but is especially useful for robotics. This is the focus of this paper.

Explicit mathematical analysis of robot controllers is implausible given the high dimensionality of the configuration space and the potential black-box representation of a learned controller. With access to an environment simulator, though, a straightforward testing approach is to roll out the robotic controller on various environments (e.g. road conditions under different weather and congestion, with or without farm animals or pedestrians, etc.), and analyze those rollouts that exhibit

5th Conference on Robot Learning (CoRL 2021), London, UK.

a specified behavior—like excessive swerving. However, with too few environments, we risk missing the condition(s) that triggers the target behavior most saliently. With too many environments, all the most salient rollouts would be close to the global maximum at the expense of diversity and coverage. For example, if a farm animal causes the most swerving, followed by a pedestrian and a dangling tree branch, using too few environments may only find the pedestrian and the tree branch while using too many would result in an exclusive focus on the farm animal. Neither case helps the human develop a correct mental model of the AV's behavior.

To address this, we introduce Robot Controller Understanding via Sampling (RoCUS), a method to enable systematic behavior inspection. RoCUS finds scenarios that are both inherently likely and elicit specified behaviors by formulating the problem as one of Bayesian posterior inference. Analyzing these scenarios and the resulting trajectories can help developers better understand the robot behaviors, and allow them to iterate on algorithm development if undesirable ones are revealed.

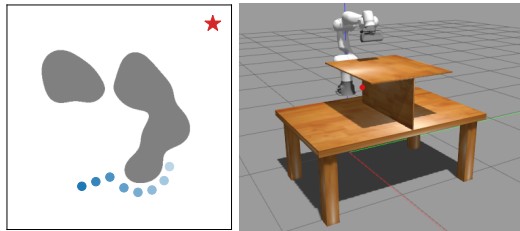

Figure 1: Two use case demos of RoCUS: 2D navigation (left) and 7DoF arm reaching (right).

We use RoCUS to analyze three controllers on two common robotics tasks (Fig. 1). For a 2D navigation problem, we consider imitation learning (IL) [6], dynamical system (DS) [7], and rapidly-exploring random tree (RRT) [8]. For a 7DoF arm reaching problem, we consider reinforcement learning (RL) [9], as well as the same DS and RRT controllers. For each problem and controller, we specify several behaviors and visualize representative scenarios and trajectories that elicit those behaviors. Through this analysis, we uncover insights that would be hard to derive analytically and thus complement our mathematical understanding of the controllers. Moreover, we include a case study on how to improve a controller based on new insights from RoCUS. As such, RoCUS is a step towards the broader goal of building more accurate human mental models and enabling holistic evaluation of robot behaviors.

## 2 Related Work

Our work lies at the intersection of efforts to understand complex model behaviors and those to benchmark robot performance. Methods to understand, interpret, and explain model behaviors are now commonplace in the machine learning community. Mitchell et al. [10] introduced Model Cards, a model analysis mechanism which breaks down model performance for data subsets. In natural language processing, Ribeiro et al. [11] introduced a checklist for holistic evaluation of model capabilities and test case generation. Booth et al. [12] introduced BAYES-TREX, a Bayesian inference framework for sampling specified classifier behaviors. In robotics, Fan et al. [13] introduced a verification framework for assessing machine behavior by sampling parameter spaces to find temporal logic-satisfying behaviors. Other efforts aim to summarize robot policies, trading off factors like brevity, diversity and completeness [14, 15]. All of these works have a shared underlying theme: treating the black box as immutable and performing downstream analyses of machine behavior [16]. RoCUS shares this theme and, similar to BAYES-TREX [12], searches for instances which exhibit target behaviors to inform accurate human mental models.

While the need for benchmarking robot performance is often expressed [17, 18, 19], these efforts usually operate on distributions of trajectories or randomly selected trajectories, and the accompanying metrics are typically task-completion based without consideration of implicit performance factors. Anderson et al. [20] put forth a recommendation of using *success weighted by path length* for navigation tasks—a task-completion metric. Cohen et al. [21] and Moll et al. [22] introduced suites of metrics for comparing motion planning approaches, and Lagriffoul et al. [23] presented a set of task and motion planning scenarios and metrics. Again, all of these proposed metrics are based solely on task completion. Lemme et al. [24] proposed a set of performance measures for reaching tasks, which are either task-completion based or require a costly human motion ground truth. Our contribution is distinct in two ways. First, we propose to sample specific trajectories which communicate controller behaviors instead of reporting metrics averaged over distributions of trajectories. Second, we introduce metrics which draw on these prior works while also including essential alternative and typically emergent quality factors, like motion jerkiness and legibility [25].

# 3  RoCUS

At a high level, RoCUS helps users understand robotic controllers via representative scenarios that exhibit various specified behaviors. It solves this by directly incorporating the distribution of scenarios, formally called *tasks*, into a Bayesian inference framework as shown in Fig. 2. A robotic problem is represented by a distribution $\pi(t)$ of individual tasks $t$. For example, a navigation problem may have $\pi(t)$ representing the distribution over target

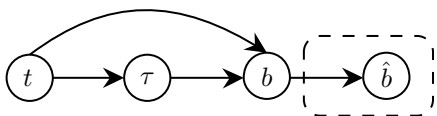

Figure 2: The graphical model for the inference problem of finding tasks $t$ and trajectories $\tau$ which exhibit specific behaviors $b$. The dashed box indicates the relaxed formulation (Eq. 2).

locations and obstacle configurations. Given a specific task $t$, the controller under study induces a distribution $p(\tau|t)$ of possible trajectories $\tau$. If both the controller and the transition dynamics are deterministic, $p(\tau|t)$ reduces to a $\delta$-function at the induced trajectory $\tau$. Stochasticity in either the controller (e.g., RRT) or the dynamics (e.g., uncertain outcome from an action) can result in $\tau$ being random. Finally, a behavior function $b(\tau, t)$ computes the behavior value of the trajectory—for example, the motion jerkiness. Some behaviors only depend on the trajectory and not the task, but we use $b(\tau, t)$ for consistency. Sec. 4 presents a list of behaviors.

The discussion on behavior in Sec. 1 is informal and implicitly combines two related but different concepts. The first concept is the behavior function $b(\tau, t)$ discussed above. The second is the specified target: for the swerving example, we are particularly interested in *maximal* behavior values. Thus, the target value can be thought of as $+\infty$. This inference problem uses the *maximal* mode of RoCUS. In other cases, we are also interested in tasks and trajectories whose behaviors *matches* a target. For example, we want to find road conditions that lead to a daily commute time of an hour, where the behavior is the travel time. This inference problem uses the *matching* mode. Since matching mode is conceptually simpler, we present it first, followed by maximal mode. The sampling procedure is the same for both modes and presented last in Alg. 1.

## 3.1  Matching Mode

The exact objective is to find tasks and trajectories that exhibit user-specified behaviors $b^*$:

$$t, \tau \sim p(t, \tau|b = b^*) \propto p(b = b^*|t, \tau)\pi(\tau|t)\pi(t). \tag{1}$$

In most cases this posterior does not admit direct sampling, and an envelope distribution is not available for rejection sampling. Markov-Chain Monte-Carlo (MCMC) sampling does not work either: since the posterior is only non-zero on a very small or even measure-zero set, a Metropolis-Hastings (MH) sampler [26] can get stuck in the zero-density region. Similar to the BAYES-TREX formulation [12], we relax it using a normal distribution formulation as shown in Fig. 2:

$$\widehat{b}|b \sim \mathcal{N}(b, \sigma^2) \quad t, \tau \sim p(t, \tau|\widehat{b} = b^*) \propto p(\widehat{b} = b^*|t, \tau)p(\tau|t)\pi(t). \tag{2}$$

This relaxed posterior is non-zero everywhere $\pi(t)$ is non-zero and provides useful guidance to an MH sampler. While $\sigma$ is a hyper-parameter in BAYES-TREX [12], we instead choose $\sigma$ such that

$$\int_{b^*-\sqrt{3}\sigma}^{b^*+\sqrt{3}\sigma} p(b)\, \mathrm{d}b = \alpha, \quad \text{with} \quad p(b) = \int_t \int_\tau p(\tau|t)\pi(t)\mathbb{1}_{b(\tau,t)=b}\, \mathrm{d}\tau\, \mathrm{d}t \tag{3}$$

being the marginal distribution of $b(\tau, t)$, which can be estimated by trajectory roll-outs. This formulation has two desirable properties. First, it is scale-invariant with respect to $b(\tau, t)$, e.g. measured under different units like meters vs. centimeters. Second, the hyper-parameter $\alpha \in [0, 1]$ has the intuitive interpretation of the approximate "volume" of posterior samples $t, \tau \mid \widehat{b} = b^*$ under the marginal $p(t, \tau) = p(\tau|t)\pi(t)$, a notion of their representativeness. Details are derived in App. A.

## 3.2  Maximal Mode

In this mode, RoCUS finds trajectories that lead to maximal behavior values: $b^* \to \pm\infty$. It can also be used for finding minimal behavior values by negating the behavior. The posterior formulation is:

$$b_0 = \frac{b - \mathbb{E}[b]}{\sqrt{\mathbb{V}[b]}}, \quad \beta = \frac{1}{1 + e^{-b_0}}, \quad \widehat{\beta} \sim \mathcal{N}\left(\beta, \sigma^2\right), \quad t, \tau \sim p(t, \tau|\widehat{\beta} = 1), \tag{4}$$

where $\mathbb{E}[b]$ and $\mathbb{V}[b]$ are the mean and variance of the marginal $p(b)$. $\sigma$ is chosen such that

$$\int_{1-\sqrt{3}\sigma}^{1} p(\beta)\, \mathrm{d}\beta = \alpha, \tag{5}$$

where $p(\beta)$ is the marginal distribution similar to Eq. 3. If $p(b)$ is normal, $p(\beta)$ is logit-normal. This formulation is again scale-invariant and has the same "volume" interpretation for $\alpha$ (App. A).

### 3.3 Posterior Sampling

The posterior sampling mechanism depends on the stochasticity of the controller and dynamics.

**Deterministic Controller & Dynamics**: When both the controller and the dynamics are deterministic, so is $\tau|t$, denoted as $\tau(t)$. Eq. 2 reduces to $t \sim p(t|\widehat{b} = b^*) \propto p(\widehat{b} = b^*|t, \tau(t))\pi(t)$, and similarly for Eq. 4.

Alg. 1 presents the MH sampling procedure. First, $\sigma$ is computed from $\alpha$ (Line 2). Then we start with an initial task $t$ (Line 3). For each of the $N$ iterations, we propose a new task $t_{\text{new}}$ according to a transition kernel and compute the forward and reverse transition probabilities $p_{\text{for}}, p_{\text{rev}}$ (Line 5). We evaluate the posteriors under $t$ and $t_{\text{new}}$ (Line 6 and 7) and calculate the acceptance probability using the MH detailed balance principle (Line 8). Finally, we accept or reject accordingly (Line 9 – 11). Note that if the proposal is rejected, the current $t$ is left unchanged *and appended to the samples*. We can discard the first $N_B$ samples as burn-in, and/or thin the samples by a factor of $N_T$ to reduce auto-correlation.

**Stochastic Controller**: When the controller and $p(\tau|t)$ are stochastic, the controller can usually be implemented by sampling a random variable $u$ (independent from $t$), and then producing the action based on the realization of $u$, as shown in Fig. 3. For instance, a Normal stochastic policy $\pi(s) \sim \mathcal{N}(\mu(s), \sigma(s)^2)$ can be implemented by first sampling $u \sim \mathcal{N}(0, 1)$ and then computing $\pi(s) = \mu(s) + u \cdot \sigma(s)$.

---

**Algorithm 1:** MH Sampling Procedure

**Input:** "Posterior volume" $\alpha$, number of samples $N$, optional burn-in $N_B$ and thinning period $N_T$.

1   samples $\leftarrow [\;]$;
2   Get $\sigma$ from $\alpha$ by Eq. 3 (matching) or 5 (maximal);
3   Randomly initialize $t$;
4   **for** $i = 1, ..., N$ **do**
5      $t_{\text{new}}, p_{\text{for}}, p_{\text{rev}} = \text{propose}(t)$
6      Get $p$ from $t$ by Eq. 2 (match) or Eq. 4 (max)
7      Get $p_{\text{new}}$ from $t_{\text{new}}$ by Eq. 2 or Eq. 4;
8      $a \leftarrow (p_{\text{new}} \cdot p_{\text{rev}})/(p \cdot p_{\text{for}})$;
9      Sample $u \sim \mathcal{U}[0, 1]$;
10      **if** $u < a$ **then**
11         $t \leftarrow t_{\text{new}}$;
12      Append $t$ to samples;
13   Optionally, discard the first $N_B$ burn-in samples and thin the samples by only keeping every $N_T$ samples;
14   **return** samples

---

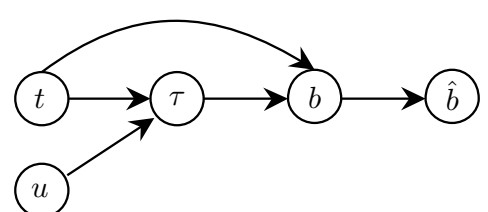

Figure 3: The same graphical model as in Fig. 2, but with the addition of stochasticity $u$ in the controller such that $\tau|t, u$ is now deterministic.

In this case, we sample in the combined $(t, \tau)$-space, with Eq. 2 being $p(t, \tau|\widehat{b} = b^*) \propto p(\widehat{b} = b^*|t, \tau(e, u))p(u)\pi(t)$, where we overload $\tau(t, u)$ to refer to the *deterministic* trajectory given the task $t$ and controller randomness $u$. It is crucial that for any $u$, we can evaluate $p(u)$. Concretely, modifying Alg. 1, $u_{\text{new}}$ is proposed alongside with $t_{\text{new}}$ (Line 5), the detailed balancing factor (Line 8) is multiplied by $p_{u,\text{rev}}/p_{u,\text{for}}$, and $t_{\text{new}}, u_{\text{new}}$ are accepted or rejected together (Line 10 – 12).

**Stochastic Dynamics**: Using the same logic, RoCUS can also accommodate dynamics stochasticity, *as long as it can be captured in a random variable $v$ and $p(v)$ can be evaluated*. We leave the details to App. B and use deterministic dynamics in our experiments.

### 3.4 The Bayesian Posterior Sampling Interpretation

RoCUS uses Bayesian sampling concepts of prior, likelihood, and posterior quite liberally. Specifically, the task distribution is defined as the prior, and thus the notion of a task being likely in the deployment context refers to high probability under the prior. Likelihood refers to the behavior saliency: how much the exhibited behavior matches the behavior specification. The act of posterior sampling then finds tasks that strike a balance between these two objectives.

The choice of explicitly modeling the task distribution is intentional, as it is not unlikely that the deployment environment will be different than the development environment. Such a domain mismatch may cause catastrophic failures, especially for learned controllers whose extrapolation behaviors are typically undefined. With a suitable task distribution, RoCUS allows more failures to surface during this testing procedure.

## 4 Behavior Taxonomy

Robot behaviors broadly belong to one of two classes: intentional and emergent. *Intentional* behaviors are those that the controller explicitly optimize with objective functions. For example, the controller for a reaching task likely optimizes to move the end-effector to the target, by setting the target as an attractor in DS, using a target-reaching objective configuration in RRT, or rewarding proximity in RL. Thus, the final distance between the end-effector and the target is an intentional behavior for all three controllers. By contrast, *emergent* behaviors are not explicitly specified in the objective. For the same reaching problem, an RL policy with reward based solely on distance may exhibit smooth trajectories for some target locations and jerky ones for others. Such behaviors may emerge due to robot kinematic structure, training stochasticity, or model inductive bias.

For trajectory $\tau$, many behavior metrics $b(\tau, t)$ can be expressed as a line integral $\int_\tau V(\mathbf{x}) \, \mathrm{d}s$ of a scalar field $V(\mathbf{x})$ along $\tau$ or its length-normalized version $\frac{1}{||\tau||} \int_\tau V(\mathbf{x}) \, \mathrm{d}s$, where $\mathrm{d}s$ is the infinitesimal segment on $\tau$ at $\mathbf{x}$ and $||\tau||$ is the trajectory length. $\mathbf{x}$ and $\tau$ can be in either joint space or task space. We introduce six behaviors: length, time derivatives (velocity, acceleration and jerk), straight-line deviation, obstacle clearance, near-obstacle velocity and motion legibility, whose mathematical expressions are in App. C. In addition, custom behaviors can also be used with RoCUS.

## 5 RoCUS Use Case Demos

In this section, we demonstrate how RoCUS can find "hidden" properties of various controllers for two common tasks, navigation and reaching. We also uncover a suboptimal controller design due to bad hyper-parameter choices, which is improved based on RoCUS insights.

### 5.1 Controller Algorithms

We consider four classes of robot controllers. The **imitation learning** (IL) controller uses expert demonstrations to learn a neural network policy which maps observations to deterministic actions. The **reinforcement learning** (RL) controller implements proximal policy gradient (PPO) [27]. While a mean and a variance is used to parameterize a PPO policy during training, the policy deterministically outputs the mean action during evaluation. The **dynamical system** (DS) controller modulates the linear controller $\mathbf{u}(\mathbf{x}) = \mathbf{x}^* - \mathbf{x}$, for the task-space target $\mathbf{x}^*$, into $\mathbf{u}_M(\mathbf{x}) = M \cdot \mathbf{u}(\mathbf{x})$ using the modulation matrix $M$ derived from obstacle configuration, as proposed by Huber et al. [7]. We give a self-contained review in App. D. The **rapidly-exploring random tree** (RRT) controller finds a configuration-space trajectory via RRT and then controls the robot through descretized segments. Notably, RRT is stochastic, and we discuss the use of controller stochasticity $u$ (c.f. Fig. 3) in App. E. The MCMC sampling uses a Gaussian drift kernel, as detailed in App. F.

### 5.2 2D Navigation Task Experiments

**Setup** In a rectangular arena with irregularly shaped obstacles, a point mass robot needs to move from the lower left to the upper right corner (Fig. 1 left). App. G details the obstacle generation and robot simulation procedures and contains more environment visualizations.

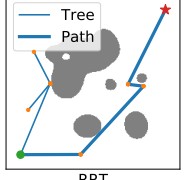
RRT

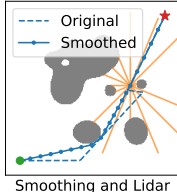
Smoothing and Lidar

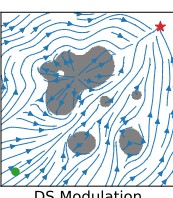
DS Modulation

Figure 4: RRT, IL and DS controllers on 2D navigation domain. Left: the RRT controller tree. Middle: smoothed RRT trajectory and lidar sensor (orange lines) for IL controller training. Right: the modulation by the DS controller.

We consider three controllers for this environment: an RRT planner, a deep learning IL policy, and a DS (Fig. 4). The RRT planner implements Alg. 2 and discretizes the path to small segments as control signals at each time step. The IL controller uses smoothed RRT trajectories as expert demonstrations, and learns to predict heading angle from its current position and lidar readings. The DS controller finds an interior reference point for each obstacle, and converts each obstacle in the environment to be star-shaped. $\Gamma$-functions are then defined for these obstacles and used to compute the modulation matrix $M$. App. H contains additional implementation details.

**Straight-Line Deviation** In most cases, the robot cannot navigate directly to the target in a straight line. Thus, the collision-avoidance behavior is a crucial aspect for navigation robots. To understand it, we sample obstacles that lead to trajectories minimally deviating from the straight line path. Since the deviation is always non-negative, we use the matching mode in Eq. 2 with target $b^* = 0$.

In Fig. 5, the top row plots posterior trajectories in orange, with prior trajectories in blue. The bottom row plots the obstacle distributions compared to the prior, with red regions being more likely to be occupied by obstacles and blue ones less likely to be obstructed.

For DS and RRT, the posterior trajectories and obstacle configurations are mostly symmetric with respect to the straight-line connection, as expected since both methods are formulated symmetrically with respect to the $x$- and $y$-coordinates. The obstacle distribution under RRT is also expected, since it seeks straight-line connections whenever possible and thus favor a "diagonal corridor" with obstacles on either side. For DS, however, obstacles are slightly *more* likely to exist at the two ends of the above-mentioned corridor. This behavior is an artifact of the DS *tail effect*, which drags the robot around the obstacle (details in App. D). By taking advantage of anchor-like obstacles at the ends of the corridor, the modulation can reliably minimize the straight-line deviation.

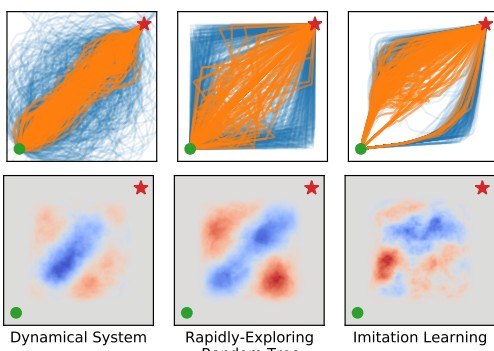

Figure 5: Top: Posterior trajectories in orange vs. prior in blue for minimal straight-line deviation behavior for three controllers. Bottom: Posterior obstacle distribution relative to the prior. Higher obstacle density regions are painted in red and lower ones in blue.

By comparison, the IL controller saliently exhibits trajectory asymmetry: it mostly takes paths on the left. It is possible that the asymmetry is due to "unlucky" samples by the MH sampler, but many independent restarts all confirm its presence, indicating that the asymmetry is inherent in the learned model. Since the neural network architecture is symmetric, we conclude that the stochasticity in the dataset generation and training procedure (e.g. initialization) leads to such imbalanced behaviors. Furthermore, the obstacle map suggests that obstacles are distributed very close to the robot path. Why does the robot seem to drive into obstacles? The answer lies in dataset generation: the smoothing procedure (Fig. 4 middle) results in most demonstrated paths navigating tightly around obstacles, and it is thus expected that the learned IL controller displays the same behavior.

**Takeaways** ROCUS reveals two unexpected phenomena. First, IL trajectories are highly asymmetric toward the left of the obstacle due to dataset and/or training imbalance. Second, both DS and IL models exhibit certain "obstacle-seeking" behaviors, the former due to the "tail-effect" and the latter due the dataset generation process. In both cases, such behavior may be undesirable in deployment due to possibly imprecise actuation, and the controller design may need to be modified. Additional studies on legibility and obstacle clearance behaviors are presented in App. I.

## 5.3 7DoF Arm Reaching Task Experiments

**Setup** A 7DoF Franka Panda arm is mounted on the side of a table with a T-shaped divider (Fig. 1 right). Starting from the same initial configuration on top of the table, it needs to reach a random location on either side under the divider. We simulate this task in PyBullet [28]. We consider three controllers: an RRT planner, a deep RL PPO agent, and a DS formulation.

RRT again implements Algorithm 2, but uses inverse kinematics (IK) to first find the joint configuration corresponding to the target location. The RL controller is a multi-layer perceptron (MLP) network trained using the PPO algorithm. The DS model outputs the end-effector trajectory in the task space, which is converted to joint space via IK, with SVM-learned obstacle definitions. App. J contains additional implementation details for each method. Overall, RRT and RL are quite successful in reaching the target while the DS is not due to the bulky robot structure, close proximity to the divider, and the task-space only modulation.

**End-Effector Movement** We find configurations that minimize the total travel distance of the end-effector for RRT and RL (DS omitted due to high failure rate). Fig. 6 (left two) shows the posterior target locations and trajectories. Notably, unlike RL, RRT trajectories are highly asymmetric, since there are straight-line connections in the configuration space from the initial pose to some target regions on the left, while every right-side goal requires at least an intermediate node.

**DS Improvement with ROCUS** Our initial DS implementation frequently fails to reach the target. This is understandable, as the DS convergence guarantee [7] is only valid in task space, in which

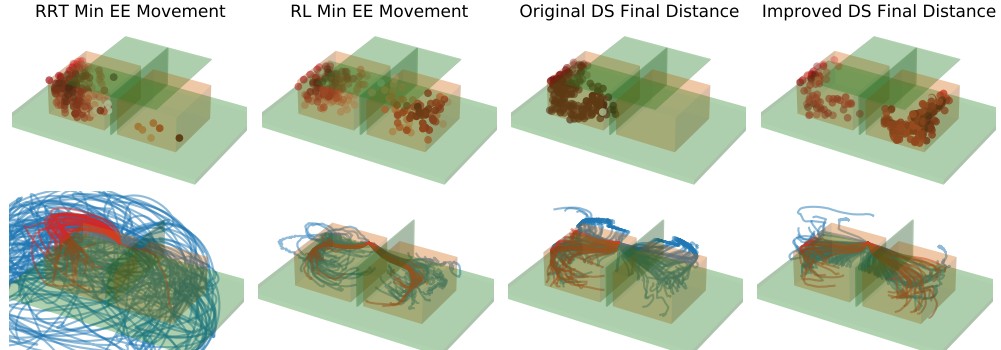

RRT Min EE Movement RL Min EE Movement Original DS Final Distance Improved DS Final Distance

Figure 6: Left: Minimal end-effector movement samples for RRT and RL. Right: Posterior samples for minimal distance from end-effector to target for the original and improved DS controllers. Top: posterior targets locations, with tabletop + divider in green and target region in orange. Bottom: posterior trajectories in red, prior trajectories in blue. Robot is mounted on the near long edge.

the modulation is defined. When the full-arm motion is solved via IK, it is possible that some body parts may collide and get stuck because of the table divider. To understand the DS behaviors, we use RoCUS to sample target locations that result in minimal final distance from the end-effector to the target (i.e., most successful executions, Fig. 6 center-right). Similar to the RRT case, the samples show strong lateral asymmetry, with all posterior target locations on the left, due to the same cause of asymmetric kinematic structure. The result points to a clear path to improve the DS controller such that it can succeed with right-side targets: increase the collision clearance of the divider so that the end-effector navigates farther away from the divider, thus also bringing the whole arm to be farther away. As detailed in App. K, this modification greatly improves the controller performance as confirmed by the new symmetry in Fig. 6 (rightmost). In addition, since the issue with DS controller mainly lies in obstacle avoidance in joint-space or on the body of the robot, additional techniques [29, 30, 31, 32] could be used and we leave them to future directions.

**Takeaway** The set of studies reveal an important implication of the robot's kinematic structure: the left side is much less "congested" with obstacles than the right side in the configuration space. While the RL controller is able to learn efficient policies for both sides, the design of certain controllers may need to explicitly consider such factors. App. K includes an additional study on legibility.

## 5.4 Quantitative Summary

We studied other additional behaviors on both tasks, and Tab. 1 summarizes prior vs. posterior mean behavior values and shows that RoCUS consistently finds samples salient in the target behavior.

| Domain | Behavior | Target | Prior (DS) | Post. (DS) | Prior (IL/RL) | Post. (IL/RL) | Prior (RRT) | Post. (RRT) |
|---|---|---|---|---|---|---|---|---|
| 2D Nav | Avg. Jerk | 0 | 1.84e-3 | 1.46e-3 | 6.95e-4 | 3.19e-4 | 4.24e-4 | 2.79e-4 |
| | Straight | 0 | 0.256 | 0.084 | 0.378 | 0.301 | 0.470 | 0.162 |
| | Legibility | min | 0.819 | 0.650 | 0.877 | 0.784 | 0.798 | 0.669 |
| | Obstacle | 0 | 0.309 | 0.229 | 0.262 | 0.218 | 0.312 | 0.241 |
| | Obstacle | max | 0.309 | 0.611 | 0.262 | 0.387 | 0.312 | 0.442 |
| Arm | Straight | 0 | 0.980 | 0.913 | 0.858 | 0.762 | 1.223 | 0.897 |
| | EE Dist | 0 | 0.934 | 0.623 | 0.958 | 0.691 | 3.741 | 1.192 |

Table 1: Quantitative results on additional tasks for two domains.

## 6 MCMC Sampling Evaluation

After confirming that RoCUS can indeed uncover significant and actionable controller insights, we evaluate the sampling procedure itself, using tasks described above as examples.

**Mixing Property** A potential downside of MCMC sampler is the slow mixing time, which causes the chain to take a long time to converge from initialization and causes consecutive samples to be highly correlated. Does this phenomenon happen for our sampling? Fig. 7 plots the behavior along the

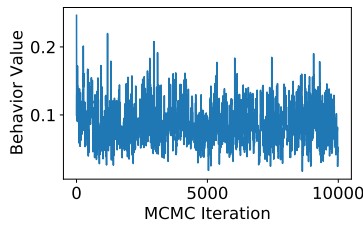

Figure 7: 2D navigation DS min straight-line deviation samples.

MCMC iterations for the DS minimal straight-line deviation behavior, showing that the chain mixes well quite fast (additional ones in Fig. 10 of App. F). Thus, a modest amount of samples, such as several thousand, is typically sufficient to model the target posterior distribution well.

**Baseline: Top-$k$ Selection**    To the best of our knowledge, RoCUS is the first work that applies the transparency-by-example formulation [12] to robotic tasks, and we are not aware of existing methods for the same purpose. Notably, adversarial perturbation algorithms [5] are *not* feasible, since stepping in simulator (or real world) is not typically differentiable. Sec. 1 discusses a straightforward alternative that runs the controller on $N$ different scenarios and pick the top-$k$ with respect to the target behavior. We demonstrate its shortcomings on the minimal straight-line deviation behavior for the 2D navigation DS controller (RoCUS samples shown in Fig. 5 left).

Fig. 8 (left) shows the trajectories of different values of $k$ for the same fixed $N$, and vice versa. While a bigger $N/k$ ratio leads to more salient behaviors in the top-$k$ samples, these examples become more concentrated around the global maximum and less diverse, making this approach especially myopic. Further, it is not easy to find the optimal $N$ to trade off between diversity and saliency of the top-$k$ samples. By contrast, RoCUS offers the intuitive $\alpha$ hyper-parameter. Fig. 8 (middle) shows that a smaller $N$ fails to highlight the "corridor" pattern while a larger $N$ makes it completely open and misses the "tail-effect anchors" at the two ends.

In addition, the hard cut-off at the $k$-th salient behavior threshold has two undesirable implications: first, every trajectory more salient than the threshold is kept but is given equal importance; second, a trajectory even slightly under the threshold is strictly discarded. By comparison, RoCUS gives more importance to more salient samples in a progressive manner, as shown in Fig. 8 right.

Finally, top-$k$ selection is very computationally inefficient. It discards all of the unselected $N - k$ samples, while RoCUS is much more efficient in that all samples after the burn-in up to the thinning factor can be kept since the posterior concentrated on the salient behavior is directly sampled.

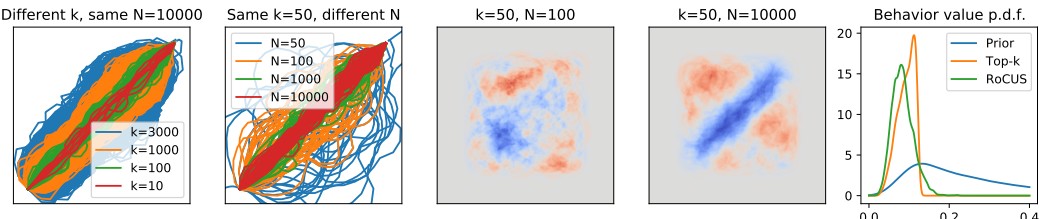

Figure 8: Top-$k$ selection baseline. Left two: trajectory distribution; middle two: obstacle distribution; right one: probability density function of behavior values.

## 7    Discussion and Future Work

RoCUS enables humans to build better mental models of robot controllers. Compared to existing evaluations on task-completion metrics for hand-designed tasks, RoCUS generates tasks and trajectories that highlight any given behavior in a principled way. We used it to uncover non-obvious insights in two domains and help with debugging and improving a controller.

While RoCUS is mainly a tool to analyze robot controllers in simulation as part of comprehensive testing before deployment, it can help understanding (anomalous) real world behaviors as well. When an anomaly is observed, RoCUS can find more samples with the anomaly for developers to identify patterns of systematic failures. Furthermore, RoCUS is not inherently limited to simulation: it only requires trajectory roll-out on specific tasks. For the arm reaching task, this is easy in the real world. For autonomous driving, "recreating" a traffic condition that involves other vehicles may be hard. However, a key feature of RoCUS is the decoupling of the task and the controller algorithm, which allows testing on simpler task variants (e.g. with props instead of real cars).

There are multiple directions for future work, including evaluation of *model updates* [33] by defining behavior functions on two controllers, better understanding the samples with explainable artificial intelligence (XAI) methods, and an appropriate interface to facilitate the two-way communication between RoCUS and end-users, as discussed in detail in App. L.

Overall, RoCUS is a framework for systematic discovery and inspection of robotic controller behaviors. We hope that the demonstrated utility of RoCUS sparks further efforts towards the development of other tools for more holistic understanding of robot controllers.

## Acknowledgement

This research is supported by the National Science Foundation (NSF) under the grant IIS-1830282. We thank the reviewers for their reviews, which are available at https://openreview.net/forum?id=5P_3bRWiRsF.

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
