# OpenReview forum: "RoCUS: Robot Controller Understanding via Sampling"
_robot-learning.org/CoRL/2021/Conference — CoRL2021 Poster_

### Official Review · Reviewer_UK3k · 2021-07-07

**Originality:** Fair
**Technical Quality:** Fair
**Clarity Of Presentation:** Good
**Impact:** 2

**Recommendation:**

Weak Reject: I recommend rejecting the paper, but will not argue for my recommendation if the majority of other reviewers have a different opinion.

**Summary:**

The paper introduces a Bayesian model for policy evaluation. They introduce in a principled way an approach to sample tasks and trajectories in which a certain desired metric, b, is maximized. To do so, the paper proposes to apply Metropolis Hasting over an approximated posterior distribution.

The paper proposes a set of behavioural metrics (trajectory length, time derivatives, straight line deviation, obstacle clearance and motion legibility) and evaluate the performance for a set of policies (Imitation Learning, Motion Planning, Reinforcement learning and Dynamic Systems) to evaluate the proposed policy evaluation method.

**Issues:**

The authours might consider adding adding qualititative metrics to improve the policy evaluation.

**Reviewer Expertise:**

Fair: Some knowledge of the area

**Strengths And Weaknesses:**

Strengths:
1. Formulating policy evaluation for robust interaction with he environment as a bayesian inference problem is interesting as it provides well-formulated problem that allow us to get deeper insights and let us compare the different methods for policy evaluation.

Weakness:
1. The main weakness of the paper is the lack of qualitative metrics. The paper proposes an evaluation method, but only evaluates the performance of a set of policies with quantitative measures.
2. RRT algorithm is well known and it would be enough to re-direct the reader to the original paper, rather than putting the algorithm itself in the paper
3. When applied the Dynamics-based approach, they first generate a task space trajectory and then map the trajectory to the configuration space by Inverse Kinematics. Instead, the performance of the DS based method would improve if it considers a multi-objective reactive motion generator in which obstacle avoidance elements can integrated [1,2,3] with the goto movements.
4. The paper lacks any information of the required amount of samples (from both trajectories and tasks) for proper evaluation.


[1] Khatib, Oussama, "Real-Time Obstacle Avoidance for Manipulators and Mobile Robots", 1986

[2] Ratliff, Nathan, et al., "Riemannian Motion Policies", 2018

[3] Urain, Julen et al., "Composable Energy Policies for Reactive Motion Generation and Reinforcement Learning", 2021

**Summary Of Recommendation:**

I suggest a weak rejection. The paper proposes a policy evaluation framework but it did not exploit it enough. The paper requires more metrics, more comparison, more policies and qualititative metrics for a better methods comparison.

---

> ### Author Response · Authors · 2021-08-25
> **Author response**
>
> Thank you for your comments. We address each weakness point in order below. Other than them, we also added a quantitative comparison against the top-k selection baseline in Sec. 6 of the revised paper. Please see our response to the meta-reviewer for more detail.
>
> 1. Originally we had some quantitative results to show the effectiveness of our sampling procedure in finding behavior-conforming examples in Table 1 and 2 of the Appendix (of the original version). Following the suggestion of Reviewer YUw7 above, we have merged and moved them to the main text as the new Table 1. We demonstrate qualitative utilities of RoCUS in the two use cases of Sec. 5, which uncovers insights of implicit model bias and leads to an improved DS controller design for the arm reaching task.
>
> 2. A similar issue is also pointed out by Reviewer YUw7, and we have delegated it to the Appendix E in the revision. We do prefer to keep it just for clarity, since the stochasticity of the RRT algorithm motivates the use of an “as-needed” kernel.
>
> 3. We thank the reviewer for the suggestion. Since our main goal is to showcase that RoCUS could be used for debugging faulty controllers, rather than proposing the best controller for each task, we add the discussion to additional ideas for controller improvement in Line 320 but leave specific implementations to future work.
>
> 4. Figure 7 of the new Sec. 6 shows the sampling trace on several tasks. We can see that the chain mixes quite well in general, and a couple thousand samples (i.e. task + induced trajectory pair) are sufficient to cover the distribution.
>
> Again, thank you for your time and efforts and please let us know if you have any additional questions or concerns.

---

### Official Review · Reviewer_YUw7 · 2021-07-23

**Originality:** Fair
**Technical Quality:** Fair
**Clarity Of Presentation:** Good
**Impact:** 2

**Recommendation:**

Weak Accept: I recommend accepting the paper, but will not argue for my recommendation if the majority of other reviewers have a different opinion.

**Summary:**

Complex robotic controllers typically demonstrate both intended and emergent behavior. Whilst intended behavior is optimized for and measured (e.g. navigate to x without collisions, grasp object y), often complex controllers exhibit emergent behaviors (e.g. jerkiness, object clearance, path length) that are also important to understand and measure. This work presents RoCUS, a method for understanding emergent controller behavior using a Bayesian sampling-based method. Unlike other controller testing techniques, RoCUS focuses on sampling representative trajectories instead of extreme or edge case trajectories that exhibit the emergent behavior of interest.


**Issues:**

* Please consider the four main issues of baseline, scope, results interpretation, and structure
* Please also address the minor points listed above.


**Reviewer Expertise:**

Fair: Some knowledge of the area

**Strengths And Weaknesses:**

**Strengths**

1. The proposed method is interesting. It appears reasonably general (provided that a function measuring the behavior of interest can be defined, and the environment scenarios of interest parameterized) and potentially a very useful way to enhance understanding of robot controllers beyond measuring task completion.
    * Of particular value is the focus on representative trajectories.
2. The experimental evidence provided is interesting (taking into account the appendix), especially Table 1 which highlights the differences between controllers. This type of controller characterization provides a rich and valuable way of understanding controllers that is complementary to task completion metrics.
3. The problem of characterizing and understanding emergent robot controller behavior is highly relevant to robot learning, especially if learning robot controllers are to be deployed in the real world. The problem setting is well motivated and explained in the paper.
4. The related work is thorough.
5. The paper is well written, the clarity of presentation is high throughout
6. The video is a valuable addition, especially the chart presented starting at 45s
7. The behavior taxonomy is valuable, well described, and appear comprehensive
    * Potentially consider looking into obstacle clearance velocity (i.e. average speed at closest distance to obstacle) in future work.
8. Sufficient detail appears to be provided to reproduce the work taking into account the appendix. The code is also released.

**Weaknesses**

No baseline
* The main alternative to RoCUS presented in the paper is a simple top-k baseline however no results using this approach are presented. This makes it difficult to assess the benefits of RoCUS over the (much simpler) baseline.

Scope
* Whilst this paper is relevant to robot learning in that it proposes a method for better understanding complex and potentially high dimensional learned robot controllers, there is no evidence provided or discussion of how the results in simulation are relevant to physical robot controllers. Potential questions of interest include:
    1. Can this method be applied to a physical robot controller? If not, why not?
    2. If this method is only applicable in simulation, can it be used to understand physical robot controllers? If not, why not and how might this limitation be overcome in the future?

The results are a little challenging to interpret.
* Figure 5 and Table 1(appendix) indicates that there is significant variation between the different controllers’ emergent behavior. This is interesting, but it is challenging to interpret and compare controllers because the likelihood of the posterior samples is not clear. Is the likelihood of the posterior samples equal across all three controllers? It would be even more useful if the likelihood could be provided for each controller and behavior for all posterior means.
* To give an example, consider Table 1(appendix), Behavior: Length. The IL and DS controllers have a similar posterior mean length (161, 166 respectively), but quite different Prior Means (161, 201 respectively). It is not clear how to compare IL and DS on trajectory length for representative samples. Are trajectories with sample length of 161 and 166 equally likely for IL and DS?

A number of the most interesting results are located in the appendix, and the main paper’s experimental evidence is not as compelling without it.
* Suggest moving Table 1 and Table 2 to the main paper.
* Suggest moving the discussion of the stochastic controller to the appendix to make space. Similarly the explanation of the RRT controller could be shortened since controller understanding instead of design is the main focus of the paper

Minor
* It is not clear why IL is used in one experimental setting (2D nav) and RL in another (Reaching)
* The sampling formulation (see eq 2) appears to be very similar to BAYES-TREX. It would be nice to see a brief discussion in the related work describing how RoCUS differs (e.g. trajectories vs classification, MCMC vs MH sampling).
* Notation: The use of No() for the Normal distribution is confusing since it is non-standard. The notation in [12] is clearer.
* Typo: 172 I think this should read (Line 7) instead of (Line 6).
* It is not clear why all the emergent behaviors are not evaluated for the 7DoF reaching task (Table 2) whilst they were for 2D nav (Table 1). It would be valuable to see Table 2 extended to include all of the behaviors from Table 1.

**Summary Of Recommendation:**

Thank you for this submission. Overall this is an interesting method that offers insight into an important problem in robot learning (understanding emergent controller behavior). The method is well explained and the results in simulation are interesting, especially when the appendix is taken into account. It is also a pleasure to read. However there are a few issues, described above, some of which are significant.

---

> ### Author Response · Authors · 2021-08-25
> **Author response**
>
> Thank you for your comments. We have added a baseline comparison with top-k selection to the new Sec. 6 of the revised paper, and discussed relationships to physical robots in Sec. 7 in blue text. For more details, we refer to the paper and our response to the meta-reviewer.
>
> For result interpretation, we do intend to make the results comparable across controllers, not with respect to specific numerical values of behavior value, but rather with respect to the overall “likelihood” of the sampled trajectories. Specifically, we introduced a single hyper-parameter, $\alpha$, defined in Eq. 3 and 5 for the matching and maximal modes, to “tune” this overall likelihood. The detailed analysis is in Appendix A of the supplementary material. We believe that this is the correct notion of comparability as it represents representativeness of the sampled tasks and the associated trajectories. With this interpretation, it is indeed that trajectories at the respective posterior means for different controllers are approximately equally likely, although the specifics depend on the exact form of the posterior, which is intractable to evaluate due to high-dimensionality.
>
> We have consolidated the two tables in appendix to a single Table 1, and pulled it into the main text under Section 5. Please see our response to the meta-reviewer for more details on the restructuring.
>
> Our response to the minor issues are in order below (all the figure numbering, etc. refer to the current revised draft):
>
> 1. We initially tried RL on the 2D navigation domain as well, and actually experimented with both a partially observable setting with simulated LiDAR input (i.e. same as the current IL input) and a fully observable setting of directly feeding in the 2D obstacle map. However, under our computational constraint of a single GPU card, we were not able to train a decent policy with both DDPG and PPO despite extensive hyper-parameter search. We suspect that this is due to the high variability of the obstacle configuration (see Fig. 11 of the Appendix for some visualization), which is prone to “local minima”. Thus, we used IL instead. For arm-reaching, PPO is successful with minimal hyper-parameter tuning, so we used its policy for evaluation.
>
> 2. Indeed, RoCUS is inspired by both the transparency-by-example principle and the posterior sampling formulation put forth by the Bayes-TrEx paper. There are several technical differences:
>
>     (a). Bayes-TrEx works on classification, while RoCUS works on robotic controllers. This motivates us to define a list of behaviors, including emergent ones.
>
>     (b). We added a maximal mode, recognizing that many robotic behavior values can be unbounded, as opposed to classification probability, which is always between 0 and 1. We proposed a reparametrization technique in Eq 4 for this mode.
>
>     (c). We proposed the hyper-parameter $\alpha$, rather than the standard deviation of the Gaussian relaxation as in Bayes-TrEx, since the $\alpha$ hyper-parameter has a much more intuitive meaning, and automatically takes care of conversion between different unit (as detailed in Appendix A).
>
> 3. We have No() to $\mathcal N()$ to refer to the normal distribution in the revision.
>
> 4. It is a typo, but actually it should be Line 5, to compute the forward and reverse transition probability $p_{for}$, $p_{rev}$. It is fixed in the revision.
>
> 5. For the arm reaching task, we found early on that the DS controller was not performing as well as expected. Thus, most efforts were devoted to find out the reason and improve its design such as the final distance between the end-effector to the target, as described in the case-study. We believe that such a concrete demonstration of controller improvement based on RoCUS analysis is more valuable than a simple compilation of behavior results, which mostly confirm the correctness of the sampling procedure itself.
>
> Again, thank you for your time and efforts and please let us know if you have any additional questions or concerns.

---

> > ### Comment · Reviewer_YUw7 · 2021-09-02
> > **Response to authors**
> >
> > Dear authors,
> >
> > Thank you for taking the time to respond to my questions and for the efforts that you made to revise the paper.
> >
> > The authors have addressed all of my concerns regarding baselines, scope, result interpretation and the paper structure. I therefore change my recommendation to weak accept.
> >
> > As I mentioned in my original review, this is an interesting method that offers insight into an important problem in robot learning. The method and results are clearly presented, and given the changes, the results are compelling.
> >
> > To further improve the paper, I would love to see the top k baseline results included in Table 1. I would be interested to know how different the Post (D) metrics using RoCUS are when compared with the top k baseline.
> >
> > I look forward to seeing the authors proposed future work of applying RoCUS to compare two controllers. It would also be interesting to see RoCUS applied to the motivating use case of finding representative scenarios for certain test cases (e.g. vehicle swerving) in the application of Autonomous Vehicles.

---

> > > ### Author Response · Authors · 2021-09-02
> > > **Thank you very much!**
> > >
> > > Dear reviewer,
> > >
> > > We are glad that our response addressed your concerns and really appreciate your support of the submission!
> > >
> > > Best,
> > >
> > > Paper authors

---

### Official Review · Reviewer_h5ZA · 2021-07-23

**Originality:** Good
**Technical Quality:** Good
**Clarity Of Presentation:** Good
**Impact:** 4

**Recommendation:**

Strong Accept: I recommend accepting the paper and will argue for my recommendation even if other reviewers hold a different opinion.

**Summary:**

In this article, the authors propose a Bayesian framework that aims to find situation in robot planning tasks that lead to trajectories which exhibit specified behaviors. For this purpose, the task distribution is defined as the prior, and the behavior saliency is the likelihood. Then, the samples of the posterior are tasks that balance these two objectives. Two simulations demonstrate the benefits of the proposed approach.

**Issues:**

- (2) What is No? I guess it's the normal distribution but the notation is uncommon and never introduced.
- Alg.1 : "numner" -> "number"
- Alg.1 : How does the choice of $N_B$ and $N_t$ affect the result?

**Reviewer Expertise:**

Fair: Some knowledge of the area

**Strengths And Weaknesses:**

Strengths:
-	Interesting and time-relevant topic as it allows to study the behavior of “black-box” controller in robotics
-	Smart idea using the Bayesian framework
- Simulation with complex examples
-	Paper is well written and structured

Weakness
-	Some minor mistakes and missing definitions
-  The baseline of the simulation seems to be too simple


**Summary Of Recommendation:**

One of the most interesting papers I have reviewed this year. The proposed approach is highly relevant as more and more "black-box" controller are used in robotics. However, the results should be compared to other more complex metrics.

Update: Based on the authors' response, I updated my recommendation to "strong accept".

---

> ### Author Response · Authors · 2021-08-25
> **Author response**
>
> Thank you for your feedback. In a certain sense, we share your concern in our limited comparison against baselines, but we actually could not identify existing approaches that try to solve the same problem, especially for robotic tasks, as discussed in detail in our response to the meta-reviewer. Thus, we hope that RoCUS could bring the importance of holistic evaluation to the attention of the community, and have it built on and extended by future work. However, we added a baseline comparison with top-k selection in Sec 6 of the revised draft, and we refer to the paper and our response to the meta-reviewer above for more discussions.
>
> No refers to the normal distribution. We have changed it to $\mathcal N$, which seems to be more standard, as requested by Reviewer YUw7.
>
> For $N_B$ and $N_T$, we found that the MCMC chain mixes pretty fast, which is studied in Fig. 7 of Sec. 6 in the revised paper (it appeared in the Appendix of the original version but several reviewers would like to see quantitative study into the sampler itself). Thus, a small burn-in sampling and thinning period would be sufficient.
>
> Again, thank you for your time and efforts and please let us know if you have any additional questions or concerns.

---

> > ### Comment · Reviewer_h5ZA · 2021-08-31
> > **Reviewer response**
> >
> > Thank you for addressing all my comments. I updated my recommendation to "strong accept".

---

> > > ### Author Response · Authors · 2021-08-31
> > > **Thank you very much**
> > >
> > > Dear reviewer,
> > >
> > > Thank you for taking the time in going over our revision and response, and we appreciate your support of the submission.
> > >
> > > Best,
> > >
> > > Paper authors

---

### Official Review · Reviewer_U7Po · 2021-07-24

**Originality:** Good
**Technical Quality:** Good
**Clarity Of Presentation:** Very Good
**Impact:** 3

**Recommendation:**

Weak Accept: I recommend accepting the paper, but will not argue for my recommendation if the majority of other reviewers have a different opinion.

**Summary:**

This work aims at providing a systematic way to debug controllers and analyze behavior saliency, in order to avoid dangerous or unexpected situations. There are various approaches to do so, as direct testing of extreme and edge cases or providing a mathematical analysis of the controller. In most cases, this is either cumbersome or extremely difficult. The focus of the paper is to establish a framework to evaluate controllers and summarize robot performance with the specific purpose of informing accurate human models. One way of testing for controller failure is to roll it out on many different scenarios, but this is a notably difficult problem, considering that it involves sampling from a posterior which is predominated by zero-density regions (salient behaviors/edge cases). They present RoCUS, a method that enables understanding of robot controllers via posterior sampling of representative behaviors.

**Issues:**

- Line 153: what do you mean when b tends to + or - infinity?
- Line 312: what do you mean with b=0?
- How does RoCUS compare to other task completion-based metrics?
- While describing figure 2 you mention that for a deterministic system and controller, the probability of a given trajectory is a delta. It is not clear to me if you are modeling also the initial condition of the dynamical system, together with the trajectory.
- Would be interesting to see more use cases of matching vs maximal mode.
- The premises of the paper are very alluring and it would be interesting to see RoCUS applied to more environments/tasks.

**Reviewer Expertise:**

Fair: Some knowledge of the area

**Strengths And Weaknesses:**

The paper is well written apart from some typos. They provide a solution to the problem of posterior sampling by relaxing the posterior and reformulate it such that one only has to tune an intuitive hyperparameter (alpha) which represents the volume of the samples. Like this, one can use methods like Metropolis-Hastings, as they do, to sample from the posterior. They also propose different alternative metrics for behavior understanding, that do not purely focus on task completion.
One of the weaknesses of the paper was a lack of a baseline comparison (with any of the methods cited in the related work for example). It could be nice to see how it improves over a less sophisticated form of sampling.

**Summary Of Recommendation:**

This paper provides a complementary approach to direct testing of extreme cases, which is very useful when the controller is deployed in situations that could hurt humans. The two presented use cases (navigation and manipulation) provided interesting insights which could be further showcased and assessed in other more complicated environments, or better, more complicated tasks, in the lack of real robotic experiments.

---

> ### Author Response · Authors · 2021-08-25
> **Author response**
>
> Thank you for your feedback. We refer to our response to the meta-reviewer above for the discussion on baseline comparison and changes to the paper draft. We address each of the issues raised. All the line and section numbers refer to the revised draft.
>
> 1. There is a slight abuse of notation here. What we meant is that we want to find tasks (e.g. obstacle configuration) that lead to excessively large (or small) values of certain behavior (e.g. total navigation distance). Since we do not have a finite target value (who knows a priori how long the navigation distance could be?), we use a reparametrization as described in Eq. 4 to sample in this formulation. Hence, the name “maximal (or minimal, if using negative behavior value) mode”.
>
> 2. We want to sample minimal straight-line deviation in this case study. From its definition in Line 520 in Appendix C, we know that it is non-negative. Thus, we choose an “anchor value” to 0 to essentially sample obstacle configurations with minimal straight-line deviation. Directly using the negation of the maximal mode (as described above) would lead to similar results.
>
> 3. We use RoCUS to refer to the entire sampling framework. It is compatible with any type of metric, defined on the trajectory. Thus, it can be used to evaluate task completion-based metrics as well. In fact, in Line 308, we used RoCUS to first evaluate the initial DS controller on minimal final distance from the end-effector to the target. This metric is arguably a task-completion distance, since we would expect successful completions to have small (or even 0) final distances. Using RoCUS, we found similar asymmetry as the previous RRT case, which led us to identify the impact of the robot kinematic structure, and led to subsequent controller improvement.
>
> 4. For the deterministic case, everything is deterministic. So if the initial condition is necessary for deterministic simulation, it needs to be fixed as well. Otherwise, the formulation in Fig 3 and Eq 7 needs to be used, as discussed in the Stochastic Dynamics section (App. B of the revised draft). Specifically for the DS controller, the trajectory is obtained by integrating the DS control law forward in time from the starting point/initial condition (x(0)=starting point). Hence, when evaluating the DS control approach we are taking into consideration the initial condition, as it is embedded in the trajectory.
>
> 5. We compiled a new Table 1 to show additional quantitative results, and also included more results in Appendix I and K, but they are more “expected” than the two use-cases highlighted in the main text, so we did not extensively discuss them.
>
> 6. Due to the space constraint, we evaluated on three distinct controllers on two of the most popular robotic problems, navigation and arm kinematics, and already demonstrated non-trivial and hard to identify issues and insights following our analysis. Thus, it would be exciting to use RoCUS on more complex tasks but we leave this to future work. We added some discussion in the revised Sec. 7 about how RoCUS can be used on physical robots and more complex tasks.
>
> Again, thank you for your time and efforts and please let us know if you have any additional questions or concerns.

---

> > ### Comment · Reviewer_U7Po · 2021-09-03
> > **Response to authors**
> >
> > I thank the authors for clarifying my doubts. I still recommend accepting the paper if the other reviewers agree.

---

> > > ### Author Response · Authors · 2021-09-03
> > > **Thank you very much**
> > >
> > > We are glad to see that our response is helpful and thank you for your support of the submission!

---

### Meta-Review · Area_Chair_spNH · 2021-08-15

**Recommendation:** Accept (Poster)
**Confidence:** 4

**Metareview:**

The paper proposes an approach to analyze the "emergent" behaviors of a controller, assuming that the behavior metric can be defined. All reviewers agree that the paper addresses an important problem in robotics.

However, to fully appreciate the contributions of the proposed method, the authors should qualitatively and quantitatively compare it against other baselines. The authors should also clearly discuss the challenges, if any, in deploying the proposed approach on real robots.

Besides addressing the major issues mentioned above, the authors should also revise the manuscript according to the other clarifications requested and the suggestions provided.

===== Post rebuttal =====

The authors have sufficiently addressed reviewers' concerns; specifically, the comparison with additional baselines was particularly insightful. I recommend an acceptance.

---

> ### Author Response · Authors · 2021-08-25
> **Author response and summary of paper revision**
>
> Thank you for your feedback. We are glad to see that the reviewers agree with us in the importance of the problem being studied. Below, we address the changes made to the revised draft that incorporate comments from all the reviewers and the meta-reviewer.
>
> We have added a baseline comparison with the top-k selection method in the new Sec. 6 of the revised draft, which highlights its various limitations and drawbacks, and also includes a visualization of the MCMC chain (i.e. original Fig 8 in Appendix) to demonstrate that the MCMC sampler converges very fast to the true posterior distribution. We refer the (meta-)reviewers to Sec. 6 of the paper for more details.
>
> We added discussions on the relationship to real world physical robots in the last section “Discussion and Future Work” of the revised paper, in blue text. In brief, we mainly intend RoCUS to be used to analyze a robot controller in simulation, as part of comprehensive testing before deployment, but it could also help understanding of real world (anomalous) behaviors.
>
> In addition, following the suggestion of Reviewer YUw7, we pulled and merged the two quantitative result tables in Appendix to the main text, as the new Table 1, and delegated discussions of stochastic dynamics (which is not used in the experiment), mathematical definitions of common behavior functions (which are widely accepted), and RRT formulation (which is again widely used) to the appendix.
>
> Finally, minor typos and grammar issues are directly fixed in the revised copy.
>
> Again, thank you for your time and efforts and please let us know if you have any additional questions or concerns.

---

### Decision · Program_Chairs · 2021-09-13

**Decision:**

Accept (Poster)

**Comment:**

The paper proposes an approach to analyze the "emergent" behaviors of a controller, assuming that the behavior metric can be defined. All reviewers agree that the paper addresses an important problem in robotics.

However, to fully appreciate the contributions of the proposed method, the authors should qualitatively and quantitatively compare it against other baselines. The authors should also clearly discuss the challenges, if any, in deploying the proposed approach on real robots.

Besides addressing the major issues mentioned above, the authors should also revise the manuscript according to the other clarifications requested and the suggestions provided.

===== Post rebuttal =====

The authors have sufficiently addressed reviewers' concerns; specifically, the comparison with additional baselines was particularly insightful. I recommend an acceptance.